# Identification of QTLs for Heat Tolerance at the Flowering Stage Using Chromosome Segment Substitution Lines in Rice

**DOI:** 10.3390/genes13122248

**Published:** 2022-11-30

**Authors:** Thanhliem Nguyen, Shijie Shen, Mengyao Cheng, Qingquan Chen

**Affiliations:** 1College of Agronomy, Anhui Agricultural University, Hefei 230036, China; 2Faculty of Natural Sciences, Quynhon University, Quynhon 590000, Binhdinh, Vietnam

**Keywords:** heat tolerance, heat sensitivity index, quantitative trait locus, seed-setting rate

## Abstract

High temperature is a major stress in rice production. Although considerable progress has been made in investigating heat tolerance (HT) in rice, the genetic basis of HT at the heading stage remains largely unknown. In this study, a novel set of chromosome segment substitution lines (CSSLs) consisting of 113 lines derived from a heat-resistant *indica* variety N22 and a heat-sensitive *indica* variety 9311 was developed and used for the analysis of the genetic basis of HT. The heat sensitivity index (HSI) calculated based on seed-setting rates under natural and high-temperature environments was used to evaluate the influence of HT at the rice heading stage. In total, five quantitative trait loci (QTLs) associated with HT were detected based on seed-setting rate (SSR) evaluation; these were named *qSSR6-1*, *qSSR7-1*, *qSSR8-1*, *qSSR9-1* and *qSSR11-1* located on chromosomes 6, 7, 8, 9 and 11, respectively. Heat-tolerant alleles of the QTLs were all derived from N22. Among them, *qSSR9-1* overlapped with QTLs identified previously, while the remaining QTLs were found novel. In particular, *qSSR7-1* explained a high phenotypic variation of 26.35% with a LOD score of 10.75, thus deserved to be further validated. These findings will increase our understanding of the genetic mechanism underlying HT and facilitate the breeding of heat-tolerant rice varieties.

## 1. Introduction

Rice (*Oryza sativa* L.) is a staple food crop for over half of the world’s population. However, the high frequency of high-temperature stress has presenteda huge challenge torice production in recent years [1]. It has been reported that the optimal temperature forrice at the seedling stage is 28–32 °C, and that for rice at the heading stage is 25–35 °C [2]. When the temperature is below 20 °C or higher than 40 °C, rice fertilization will be seriously harmed [3]. Moreover, high temperature has been found to cause heat stress for rice growth, which led to the reduction in rice yields [4]. Therefore, it is becoming urgent to breed heat-tolerant rice varieties or identify them from pre-existing germplasms.

Many studies have primarily focused on HT in rice at the flowering stage or booting stage. For instance, high temperatures were found to induce sterility in rice at flowering [5]. It was also reported that high temperature (>33.7 °C) affected spikelet fertility of rice [6]. Following the increased temperature and the prolonged duration, the SSR gradually decreased [7]. Additionally, many HT-related QTLs have been detected on all 12 chromosomes in rice, including *qHTSF1.1*, *qHTSF1.2*, *qHTS1a*, *qHTB1-1*, *qHTS1b*, *qHTB2*, *qHTSF2.1*, *qHTSF3.1*, *qHTB3-1*, *qHTB3-2*, *qHTB3-3*, *qHTS3*, *qHTS4*, *qHTSF4.1*, *qHTSF4.2*, *qHTB4-1*, *qHTB4-2*, *qHTSF5.1*, *qHTB5-1*, *qHTB5-2*, *qHTB6*, *qHTSF6.1*, *qHTSF7.1*, *qPSL^ht^7*, *qDFT8*, *qSTIPSS9.1*, *qHTSF9.1*, *qHTB10-1*, *qHTB10-2*, *qHTB11*, *qHTSF11.2* and *qSTIPSS12.1* [8,9,10,11,12,13,14,15]. In particular, one major QTL (*qHTSF4.1*) has been observed in several studies across different genetic backgrounds, and it was validated to increase spikelet fertility under heat stress at flowering [9,16], indicating that this QTL is a valuable target for enhancing HT in rice at the flowering stage. Among the detected QTLs, only a few are fine-mapped and isolated. The first to be fine-mapped was *qHTB3-3* on chromosome 3, which mapped between the markers RM3525 and 3-M95, approximately 2.8 Mb [11]. The second, *qHTB1-1*, was fine mapped within a 47.1 kb region between markers RM11633 and RM11642, using homozygous recombinants screened from larger BC_6_F_2_ and BC_6_F_3_ populations [15]. More recently, *qHTT8* on chromosome 8, which is associated with abiotic stress tolerance, was mapped into the 3,555,000–4,520,000 bp and delimited to a 0.965 Mb region containing 65 predicted genes and 10 putative predicted genes [17].

Several rice populations have been used in QTL analysis, such as backcross inbred lines (BILs) [18], recombinant inbred lines (RILs) [19] and CSSLs [20]. Of which, CSSLs are generated by crossing a donor parent with a recipient, then followed by several backcrosses to the recurrent parent. The CSSLs contain the entire genome of the donor parent based on marker-assisted selection (MAS). Each line of CSSLs carries only one or a few homozygous fragments of the donor genotype in the genetic background of the recurrent parent, thereby eliminating genetic background noise and allowing the detection of QTL with additive minor effects [21,22]. Currently, this strategy has been widely applied in rice as a powerful tool to precisely detect QTLs of important agronomical traits without concerning complex interactions among QTLs [12,23,24].

In this study, a novel set of the CSSLs population was developed by using MAS with the heat-tolerant cultivar N22 as the donor and the heat-sensitive cultivar 9311 (an elite variety with good agronomic traits and available genome sequences) as the receptor parent. Subsequently, this set of population was applied to identify QTLs associated with HT assessed by SSR at the heading stage. The results will provide a useful clue to understand the genetic basis of HT in rice and contribute to breeding heat-tolerant rice varieties in the future.

## 2. Materials and Methods

### 2.1. Plant Materials and Cultivation

In this study, we used heat and drought tolerant variety N22 to improve heat tolerance in 9311 which has weak resistance to heat stress. The donor parent N22 released by The International Rice Research Institute (IRRI) was recognized as a variety with specific heat resistance and drought tolerance [25], but poor agronomic traits, thus it is very difficult to be directly applied in production. The receptor parent 9311 collected from Institute of Agricultural Sciences of Lixiahe District in Jiangsu Province was an elite variety which belongs to the rice genome sequencing project [26], and has good agronomic traits and wide compatibility; however, it is very sensitive to high temperature.

The two parental cultivars (9311 and N22) and a set of 113 CSSLs were sown at Teaching and Research Center of Anhui Agriculture University, Anhui Province and at Lingshui County in Hainan Province from 2009. Seedlings were transplanted in rows of 10 plants with an intra-row spacing of 13 cm and inter-row spacing of 15 cm, and managed following standard practices. All local, national or international guidelines and legislation were adhered to in the production of this study.

### 2.2. Construction of CSSLs

The F_1_ plants derived from a cross between 9311 and N22 were backcrossed to the recurrent parent 9311 to produce 98 BC_1_F_1_ plants. Then, BC_1_F_1_ generation was preliminarily detected to produce BC_2_F_1,_ using 143 simple-sequence-repeat markers distributed across the 12 rice chromosomes. BC_2_F_1_ plants were then backcrossed twice to 9311, to produce a BC_4_F_1_ generation. According to the genotype determination, the BC_4_F_1_ generation was selfed consecutively to generate BC_4_F_2_, BC_4_F_3_, and BC_4_F_4_, and finally a set of CSSLs consisted of 113 lines was developed to nearly cover the entire N22 genome.

### 2.3. DNA Isolation and PCR

The DNA was extracted from freshly frozen leaves of individuals by using improved CTAB method [27]. Extracted DNA was stored in ultra-pure water at −20 °C in a refrigerator. Simple-sequence-repeat marker primers were selected from genetic marker maps and public databases, and synthesized by Sangon Biotech Company (Shanghai, China). DNA amplification was performed using PCR with the following conditions: 94 °C for 4 min; 33 cycles of 94 °C for 15 s, 55 °C for 15 s, and 72 °C for 30 s; and a final cycle of 72 °C for 5 min. PCR products were separated in 4% polyacrylamide denaturing gels (PAGE), and bands were visualized using the silver-staining protocol [28].

### 2.4. Evaluation of Heat Tolerance Traits

A total of 113 CSSLs and parents were collected and used for heat resistance evaluation in 2014 and 2015 at the experimental farm of Teaching and Research Center, Anhui Agriculture University. Seed-settingrate was taken as the indicator of HT and used for QTL analysis.

For the SSR and HIS measurement, CSSLs and the two parents were planted under two different conditions when they were grown to the four-leaf stage. The plants grown under natural environment condition of the field were considered as the control, while those planted in pots and moved into a greenhouse for high-temperature treatment were the treatment. The temperature was set to 38 ± 2 °C and the humidity was 75 ± 5% from 8:00 to 16:00 in the greenhouse, while during the rest periods of a day, the greenhouse films were opened to make sure the growth condition of the greenhouse was the same as that of the natural environment. High-temperature treatment was continuously performed until the end of the heading stage, then these plants were moved back to the natural environment until maturity. Finally, five panicles of each line were randomly harvested to calculate the SSR. The mean SSR of the five panicles selected in each pot was investigated as the HSI and used to evaluate HT of the CSSLs and analyze genetic effects of the substitution fragments. SSR and HSI were calculated according to the following formulas:$$\mathrm{SSR}\;\left(\%\right)=\frac{\mathrm{The}\;\mathrm{number}\;\mathrm{of}\;\mathrm{fully}\;\mathrm{filled}\;\mathrm{seeds}}{\left(\mathrm{The}\;\mathrm{number}\;\mathrm{of}\;\mathrm{fully}\;\mathrm{filled}\;\mathrm{seeds}+\mathrm{The}\;\mathrm{number}\;\mathrm{of}\;\mathrm{empty}\;\mathrm{seeds}\right)}\;\times \;100\%$$
$$\mathrm{HSI}\;\left(\%\right)=\left(\mathrm{Seed}\;\mathrm{setting}\;\mathrm{rate}\;\mathrm{of}\;\mathrm{control}\;–\mathrm{Seed}\;\mathrm{setting}\;\mathrm{rate}\;\mathrm{of}\;\mathrm{heat}\;\mathrm{stress}\right)$$

### 2.5. Statistical Analysis and QTL Mapping

Statistical analyses were conducted using SPSS 20.0 software (SPSS, Chicago, IL, USA). QTL analyses were performed using QTLIciMapping 4.2 (http://qtl-icimapping.software.informer.com/, accessed on 12 October 2014), and recombination values were converted to cent-Morgans (cM) using the Kosambi mapping function. A total of 143 simple-sequence-repeat markers were used in construction of a linkage map. IciMapping 4.2 was also used to calculate the phenotypic variance explained (PVE, %) by a QTL and the additive effect. Primer 5.0 software was used to develop simple-sequence-repeat markers for fine mapping. The genomic sequence was obtained from the National Center for Biotechnology Information (NCBI) (http://www.ncbi.nlm.nih.gov/, accessed on 16 May 2010). RAP-DB (http://rapdb.dna.affrc.go.jp/, accessed on 19 August 2013) was used to carry out gene annotation within specific genomic regions according to the Nipponbare genome sequence. The GGT 2.0 software (http://www.plantbreeding.wur.nl, accessed on 8 May 2015) was used to calculate pair-wise *r*^2^ values between 143 markers distributed throughout the genome and analyze genotype of each sample and distribution of introgressed segments [29]. The QTLs were named as follows: the abbreviation of the corresponding trait (e.g., HT for heat tolerance and SSR for seed-setting rate) followed by 1–12 (the rice chromosome on which the corresponding QTL was detected), and a final number indicating the marker interval on one chromosome. For example, *qSSR6-1* indicates that the 1st interval for seed-setting rate that was detected on chromosome 6.

## 3. Results

### 3.1. Polymorphic Simple-Sequence-Repeat Markers Identification

In total, 356 simple-sequence-repeat markers distributed throughout the 12 rice chromosomes were applied to analyze polymorphisms between 9311 and N22, among which 143 (40.17%) were polymorphic between these two parents (Table 1, Figure 1). The average distance between two adjacent markers on the rice linkage map was 11.3 cM. The number of polymorphic markers in each chromosome ranged from 9 to 18. The polymorphism ratio of chromosome 3 was as high as 52.94% and that of chromosome 12 was as low as 32.14%. Additionally, the polymorphism ratio of other chromosomes ranged from 33.33% to 48.39%. Subsequently, these polymorphic simple-sequence-repeat markers were used to develop CSSLs.

### 3.2. CSSLs Development and Marker-Assisted Selection (MAS)

CSSLs were schematically developed according to the procedure of Figure 2. Following the initial cross between 9311 and N22, 9311 was selected as the recurrent parent to backcross with the hybrid. A total of 98, 110, and 121 individuals from BC_1_F_1_, BC_2_F_1_, and BC_3_F_1_ populations were genotyped, respectively, using the 143 polymorphic simple-sequence-repeat markers. From each backcross generation, the optimal individuals that have minimal number of donor segments were selected to make sure that all the selected lines can nearly cover the whole genome of the donor parent. In winter 2011, a total of 129 BC_4_F_1_ individuals were obtained and planted for self-fertilization to produce BC_4_F_2_. After screening of 2580 BC_4_F_2_, 179 individuals were selected and then planted for self-fertilization to produce BC_4_F_3_. Similarly, 127 out of 3580 BC_4_F_3_ individuals were selected and self-fertilized to produce BC_4_F_4_. Finally, a set of CSSLs including a total of 113 lines (BC_4_F_4_) was developed to covernearly the entire N22 genome.

### 3.3. Distribution of Substituted Segments on Chromosomes in CSSLs

There were 166 homozygous chromosome segments and 9 heterozygous segments introgressed in the 113 CSSLs, on average, each CSSL contained 1.6 introgressed chromosome segments (Table 2, Figure 3). Moreover, the distribution of introgressed chromosome segments was uneven among the 12 chromosomes. Chromosome 3 had the most introgressed segments (20), while chromosome 1, 11 and 12 had the fewest segments (11). Among the 113 CSSLs, the length of the 175 introgressed chromosome segments ranged from 0.5 cM to 59 cM, with an average of 15.3 cM (Table 2, Appendix A). Specifically, 36.6% of the substituted segments were smaller than 10 cM, 34.3% were from 10 to 20 cM, 20% ranged from 20 to 30 cM, and 12.1% were over 30 cM (Appendix A).

### 3.4. Phenotypic Performance of SSR under Natural and High Temperature at the Heading Stage

Seed-setting rates under natural and high temperature at the heading stage were used to calculate the HSI to assess HT. If HSI was low, HT would be good and vice versa. As shown in Table 3, the SSRs of 9311 in natural environment (E1, 78.84% and 93.88%) were significantly higher than that in the high-temperature environment (E2, 3.14% and 11.03%), and the HSI values of the parental varieties 9311 were 75.70% and 82.85%, respectively. Similarly, the SSRs of N22 in natural environment (E1, 97.48% and 97.52%) were also higher than that in the high-temperature environment (E2, 50.21% and 61.89%), and the HSI values of the parental varieties 9311 were 47.27% and 35.63%, respectively. The results showed that the HT of N22 was better than that of 9311, indicating that cultivar N22 was more tolerant to heat stress than cultivar 9311. In the natural environment, the SSRs of CSSLs ranged from 35.37 to 85.21% with a mean of 71.38% in 2014, and from 68.76 to 94.69% with a mean of 89.6% in 2015, respectively. In the high-temperature environment, however, the SSRs of CSSLs ranged from 1.03 to 36.52% with a mean of 8.56% in 2014, and ranged from 2.1 to 52.39% with a mean of 18.19% in 2015, respectively. Thus, the mean HSI of the CSSLs was 62.74% in 2014 and 71.31% in 2015. The CSSL population segregation for the SSR of rice was distributed continuously in the two environments (Appendix A). These results strongly indicated that high temperature at the heading stage significantly hindered rice production.

### 3.5. QTL Mapping for the SSR under High Temperature at the Heading Stage

In total, five QTLs (*qSSR6-1*, *qSSR7-1*, *qSSR8-1*, *qSSR9-1*, and *qSSR11-1*), referred to the HSI under high temperature at the heading stage, were detected and mapped on chromosomes 6, 7, 8, 9 and 11, respectively (Table 4, Figure 4). The LOD values ranged from 2.81 to 10.75, and the PVE ranged from 5.83 to 26.35%. Among them, *qSSR7-1* (RM248) had a high PVE of 26.35% with a LOD score of 10.75, therefore was considered as a major QTL for HSI under high temperature. *qSSR11-1* (RM224) had a medium PVE of 14.21%, while the remaining three had low PVEs which ranged from 5.83% to 7.66%. The additive effects of these QTLs ranged from −8.17 to −17.37, indicating that these QTLs contributed by N22 had synergistic effect on the HSI of rice.

## 4. Discussion

High temperature is one of the major environmental factors influencing rice growth and productivity. It is observed that 42 °C high temperature affects the early growth stage (germination and seedling) in rice [30]. In addition, high temperature caused low SSR and reduced yield especially during the flowering period. One of the main reasons is that heat stress can cause bad anther dehiscence, which resulted in low pollen germination of the stigma and reduced pollen production and spikelet fertility [31,32]. Therefore, spikelet fertility and SSR were commonly used as indicators of HT in rice [9,11,12]. In this study, SSR was selected for HT evaluation and QTL mapping at the rice heading stage.

We identified five detected QTLs (*qSSR6-1*, *qSSR7-1*, *qSSR8-1*, *qSSR9-1* and *qSSR11-1*) associated with HT (Figure 1, Table 4). Among them, *qSSR9-1* was identified on chromosome 9 and accounted for 7.66% of the phenotypic variations. Similarly, a QTL (*qSTIPSS9.1*) for HT in rice was identified on chromosome 9 through using a 5K SNP array [10]. It is worthy to note that two HT QTLs, *qHt9a* (RM108-RM242) and *qHt9a* (RM242-RM566), which shared the same location of *qSSR9-1*, were previously reported using a set of RILs [33], indicating that *qSSR9-1* is a major QTL for the HT of rice. *qSSR8-1* located on chromosome 8 and accounted for 6.11% of the phenotypic variations. Two QTLs, *qDFT8* and *qHT-8*, were also previously identified for HT in rice and located on chromosome 8, which explained 31.10% and 51.67% of the phenotypic variation, respectively [12,34]. A recent study found a HT QTLs, *qHTT8,* at the anthesis of rice located between 3,555,000 and 4,520,000 bp on chromosome 8 by using BSA combined with WGS [18], but they were distinctly different from *qSSR8-1* based on their mapping results, suggesting that *qSSR8-1* could be a novel QTL for HT. *qSSR7-1* was found to be a major QTL located on chromosome 7, which explained up to 26.35% of the phenotypic variance. *qPSL^ht^7*, which was associated with spikelet fertility under high temperature in “Sasanishiki”/“Habataki” CSSLs population across three environments, was located adjacent to *qSSR7-1* on chromosome 7 and explained 79.0% of the phenotypic variation [12]. In addition, one QTL (*qHTSF7.1*) had also been identified on chromosome 7 but overlapped with *qPSL^ht^7* [13], which indicates that *qSSR7-1* might also be a new QTL. *qSSR6-1* and *qSSR11-1* explained 5.83% and 14.21% of the phenotypic variation, respectively. On the basis of data obtained from the QTL Annotation Rice Online Database [Q-TARO, http://qtaro.abr.affrc.go.jp/, accessed on 28 November 2021] and comparison with previously reported QTLs, the two minor QTLs (*qSSR6-1*, *qSSR11-1*) detected in this study might be new.

Since extremely high temperatures caused significant loss in rice production, breeding heat-tolerant rice varieties or identifying heat-tolerant rice varieties from pre-existing germplasm has become a big concern to rice breeders.The Early Morning Flowering (EMF) trait from *Oryza glaberrima* was used to screen heat tolerance inimproved and traditional rice varieties [35]. Compared with EMF trait screen, the progeny selection of heat resistant plants in a traditional crossing program requires high labor and economic costs. Therefore, the MAS breeding has become more and more essential for breeders to breed heat-tolerant rice varieties [12]. In recent years, many putative QTLs for HT have been identified in rice; however, the QTLs of stable effect still remain rare. Hence, we designed this research to explore and further confirm useful QTLs associated with HT of rice in heading stages. Of the five QTLs identified in this study, *qSSR9-1* is consistently identified with previous studies [9,33], and can be used for rice HT improvement by MAS, while the others are novel and need to be further confirmed. These findings would contribute to better understanding of the genetic basis of HT in rice and accelerating the process of breeding heat-tolerant rice varieties.

## 5. Conclusions

We used N22 (with strong resistance to heat stress) to improve heat tolerance in 9311 (with low heat tolerance). Heat sensitivity index was used as the major criterion to evaluate heat tolerance. *qSSR7-1*, a major QTL for heat tolerance, was located on chromosome 7 and explained a high phenotypic variation of 26.35% with a LOD score of 10.75. Thus, fine mapping and cloning of this QTL is required in future work in breeding for resistance to heat stress in rice.

## Figures and Tables

**Figure 1 genes-13-02248-f001:**
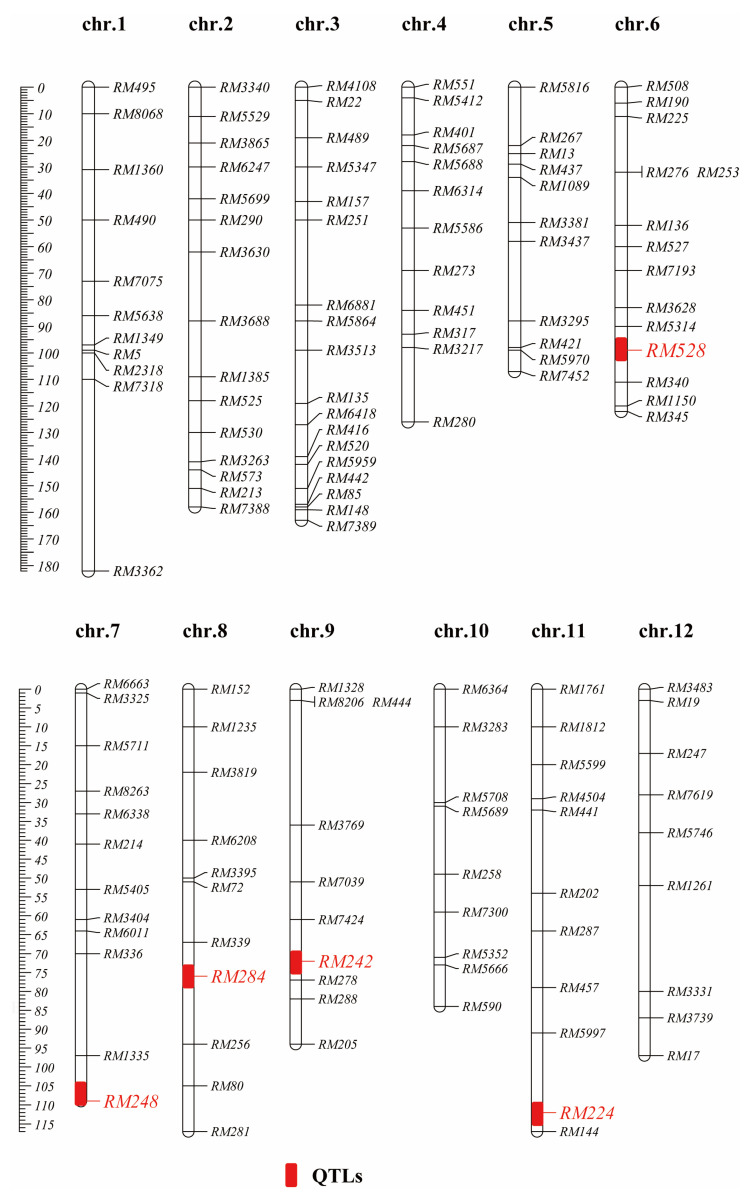
Genetic location of 143 polymorphic simple-sequence-repeat markers and distribution of QTLs for HT traits using simple-sequence-repeat. Molecular markers are shown to the right of chromosomes, and genetic locations (cM) of each marker shown to the left of chromosomes. Red rectangles indicate the QTLs for HT.

**Figure 2 genes-13-02248-f002:**
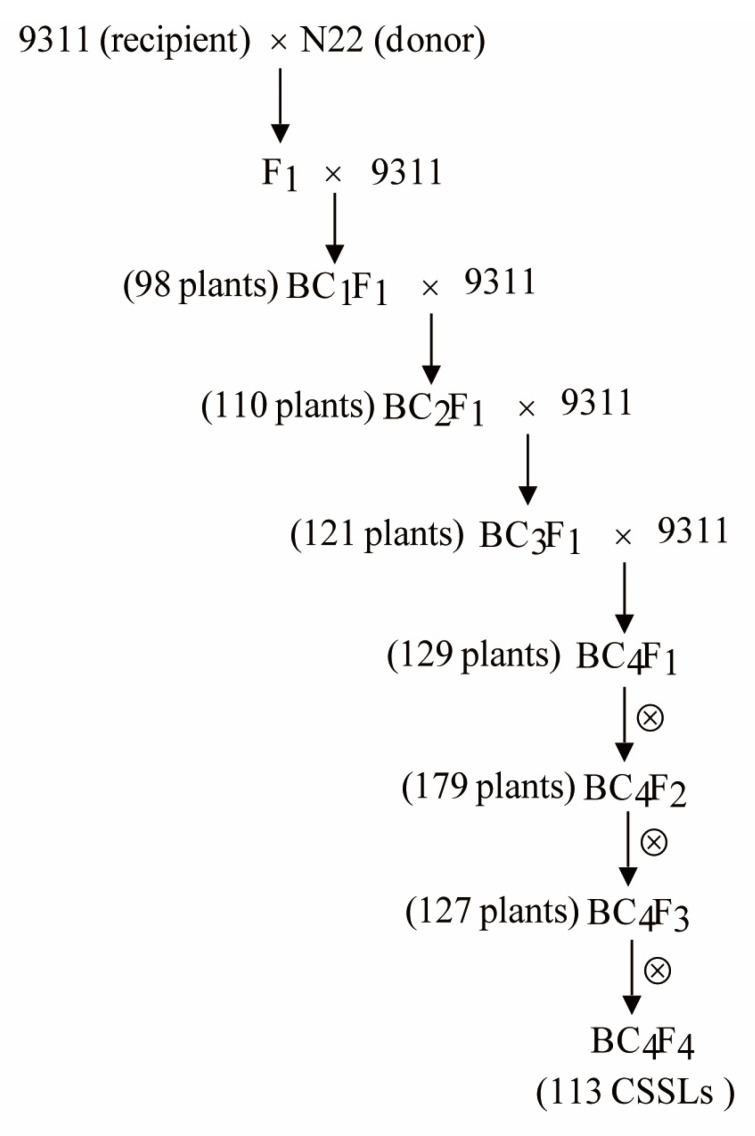
Schematic of the development of CSSLs carrying N22 chromosome segments in 9311 genetic background.

**Figure 3 genes-13-02248-f003:**
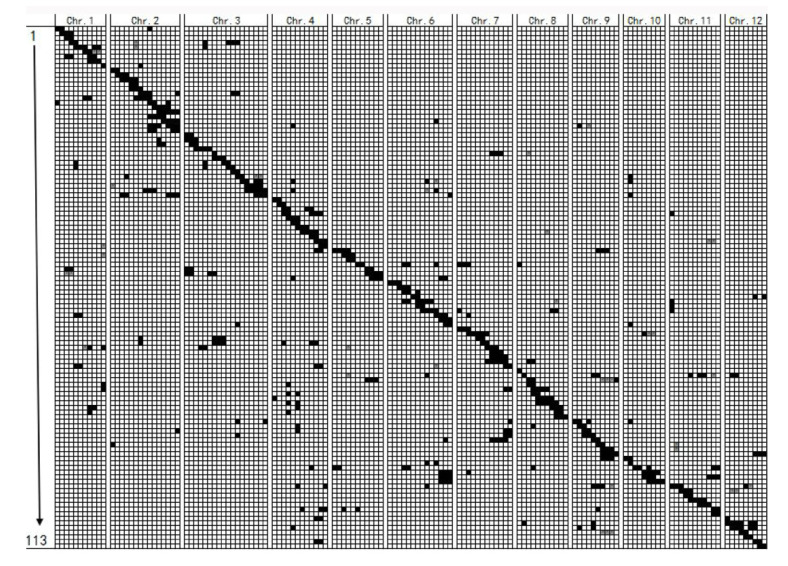
Graphic of genotypes of the 113 CSSLs. Black regions indicated homozygous segments, while gray regions indicate heterozygous segments. Regions with a white background represent 9311 background. The horizontal axis indicates marker segments on chromosome 1 to 12, and the vertical axis indicates CSSLs.

**Figure 4 genes-13-02248-f004:**
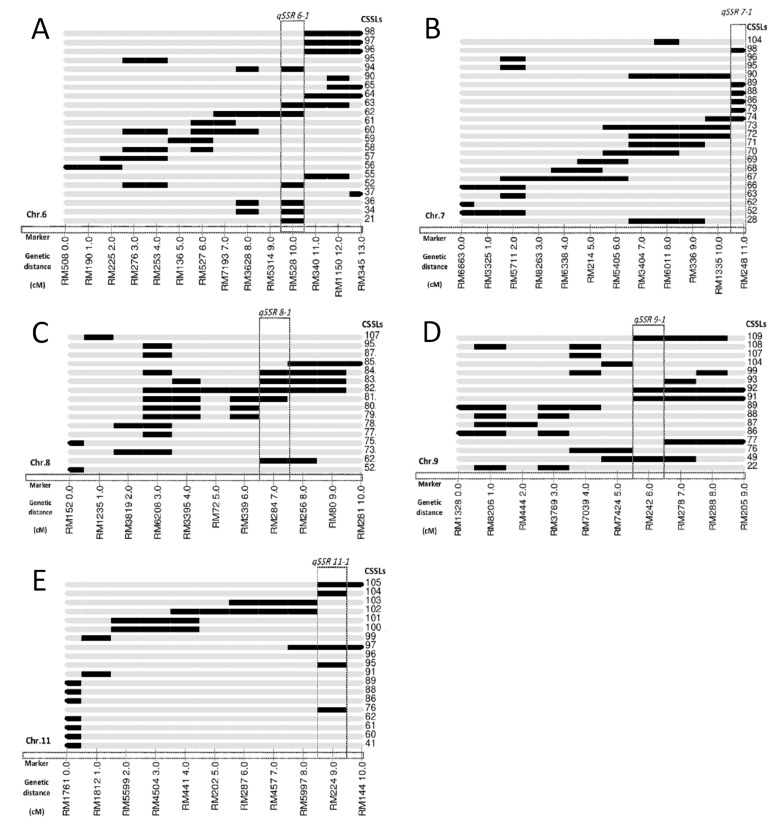
Substitution mapping of *qSSR6-1* (**A**), *qSSR7-1* (**B**), *qSSR8-1* (**C**), *qSSR9-1* (**D**) and *qSSR11-1* (**E**) for SSR on rice chromosomes.

**Table 1 genes-13-02248-t001:** Distribution of polymorphic markers on the 12 chromosomes.

Chromosome	Number of Polymorphic Marker	Number of Total Marker	Polymorphic Rate
Chr.1	11	33	33.33%
Chr.2	15	31	48.39%
Chr.3	18	34	52.94%
Chr.4	12	31	38.71%
Chr.5	11	28	39.29%
Chr.6	14	33	42.42%
Chr.7	12	31	38.71%
Chr.8	11	29	37.93%
Chr.9	10	23	43.48%
Chr.10	9	25	36.00%
Chr.11	11	30	36.67%
Chr.12	9	28	32.14%
Average	11.9	29.7	40.17%
Total	143	356	

**Table 2 genes-13-02248-t002:** Substitution of N22 segments in CSSLs.

Chromosome	Number of Lines	Number of Segments	Total Segment Length (cm)	Average Segment Length (cm)
Chr.1	9	11	167.7	15.2
Chr.2	14	19	370.0	19.5
Chr.3	14	20	238.5	11.9
Chr.4	11	14	233.6	16.7
Chr.5	7	17	226.4	13.3
Chr.6	10	14	187.5	13.4
Chr.7	9	17	276.9	16.3
Chr.8	11	16	269.7	16.9
Chr.9	8	12	187.8	15.7
Chr.10	6	13	178.2	13.7
Chr.11	7	11	181.8	16.5
Chr.12	7	11	164.8	15.0
Total	113	175	2682.9	15.3 ^a^

^a^ Average length of all introgression segments.

**Table 3 genes-13-02248-t003:** Phenotype data of seed-setting rate of rice under natural and high temperature for CSSLs and parents (9311 and N22) across two environments.

Year	Materials	Environment ^a^	SSR (%)			HSI (%)			*p*-Value ^b^
			Mean	Max	Min	Mean	Max	Min	
2014	9311	E1	78.84	-	-	75.70	-	-	0.0001 **
		E2	3.14	-	-				
	N22	E1	97.48	-		47.27	-	-	0.0054 **
		E2	50.21	-	-				
	CSSLs	E1	71.38	85.21	35.37	62.74	79.20	31.13	-
		E2	8.56	36.52	1.03				
2015	9311	E1	93.88	-	-	82.85	-	-	0.0001 **
		E2	11.03	-	-				
	N22	E1	97.52	-	-	35.63	-	-	0.0076 **
		E2	61.89	-	-				
	CSSLs	E1	89.60	94.69	68.76	71.31	90.50	18.45	-
		E2	18.19	52.39	2.10				

Note: ^a^ E1 indicates natural environment, while E2 indicates high-temperature environment; ^b,^** means the significance levels of 0.01 between 9311 and N22.

**Table 4 genes-13-02248-t004:** QTLs for seed-setting rate of rice under high temperature detected in CSSLs population.

QTLs	Chromosome	Marker	LOD Value	PVE (%)	Additive Effect
*qSSR6-1*	6	RM528	2.81	5.83	−8.17
*qSSR7-1*	7	RM248	10.75	26.35	−17.37
*qSSR8-1*	8	RM284	2.95	6.11	−9.12
*qSSR9-1*	9	RM242	3.63	7.66	−11.36
*qSSR11-1*	11	RM224	6.37	14.21	−13.90

## Data Availability

The data supporting the findings of this study are included in the Appendix A .

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
