# Peer review of "Identification of QTLs for Heat Tolerance at the Flowering Stage Using Chromosome Segment Substitution Lines in Rice"

_genes, 2022, doi:10.3390/genes13122248_

Round 1
Reviewer 1 Report
The authors of this article described the identification of QTLs associated with heat tolerance at the flowering stage using chromosome segment substitution lines (CSSLs) in rice. They crossed a heat and drought-tolerant variety with poor agronomic traits N22 with 9311 which has good agronomic characteristics but is sensitive to higher temperatures. Through a series of backcrossing progenies with 9311, they developed 113 CSSLs. They further evaluated them for heat tolerance by scoring seed setting rate and heat sensitivity index and also screened the population using 143 polymorphic simple sequence repeat markers. Through QTL mapping, they were able to detect five QTLs each on chromosomes 6, 7, 8, 9, and 11 referred to as the HSI under high temperature at the heading stage. The QTL on chromosome 7 had the highest PVE of 26.35% with a LOD score of 10.75. This QTL can be utilized in the future in the heat tolerance breeding of rice varieties.
The background, materials, and methods are well described and the results are aptly summarized and discussed in this manuscript.
Author Response
Thank you so much for your review.

Reviewer 2 Report
The authors performed a good breeding program by developing the breeding population and doing QTL mapping. The work was big and valuable. However, the manuscript has some concerns. The concerns were presented as comments in the reviewed MS. Please find the attached file.

Author Response
Thank you so much for your review.
